# A Forest Point Cloud Real-Time Reconstruction Method with Single-Line Lidar Based on Visual–IMU Fusion

**Chunhe Hu** [1,2], **Chenxiang Yang** [1,2], **Kai Li** [1,2] **and Junguo Zhang** [1,2,*]

1    School of Technology, Beijing Forestry University, Beijing 100083, China; huchunhe@bjfu.edu.cn (C.H.);
     yang_chen_xiang@bjfu.edu.cn (C.Y.); likai@bjfu.edu.cn (K.L.)
2    Research Center for Biodiversity Intelligent Monitoring, Beijing Forestry University, Beijing 100083, China
*    Correspondence: zhangjunguo@bjfu.edu.cn; Tel.: +86-10-6233-6398

**Featured Application: The research results of this paper are mainly applied to a 3D reconstruction of a forest with a single-line lidar, obtaining tree distribution at low cost.**

**Abstract:** In order to accurately obtain tree growth information from a forest at low cost, this paper proposes a forest point cloud real-time reconstruction method with a single-line lidar based on visual–IMU fusion. We build a collection device based on a monocular camera, inertial measurement unit (IMU), and single-line lidar. Firstly, pose information is obtained using the nonlinear optimization real-time location method. Then, lidar data are projected to the world coordinates and interpolated to form a dense spatial point cloud. Finally, an incremental iterative point cloud loopback detection algorithm based on visual key frames is utilized to optimize the global point cloud and further improve precision. Experiments are conducted in a real forest. Compared with a reconstruction based on the Kalman filter, the root mean square error of the point cloud map decreases by 4.65%, and the time of each frame is 903 μs; therefore, the proposed method can realize real-time scene reconstruction in large-scale forests.

**Keywords:** single-line lidar; visual–IMU fusion; nonlinear optimization algorithm; point cloud reconstruction

## 1. Introduction

High-precision sensors can collect information under forest canopies concerning the morphological structure, key characteristics, and distribution of trees. They can provide an important scientific basis for revealing the growth law of trees. Among the most commonly used sensors, lidar has received extensive attention in the field of forestry monitoring equipment due to its good stability and high precision, and it is little affected by the surrounding environment [1–6].

A multi-beam lidar is mainly applied to 3D reconstruct a space. We can obtain 6-dof pose and more spatial environment information based on multi-beam laser scanning. In 2014, Zhang et al. [7] proposed the LOAM framework, which uses the iterative closest point (ICP) algorithm to realize point cloud matching. It divides localization and mapping into two different parts. First, the pose is roughly estimated according to the high-frequency odometry, and then, through point cloud data registration and matching based on ICP, the pose is optimized, and a 3D point cloud map is constructed. This is the first simultaneous localization and mapping (SLAM) framework for 3D point cloud mapping, but due to the lack of back-end optimization, the drift is large. Shan et al. [8] added back-end optimization on the base of LOAM and proposed a LeGO-LOAM framework that can be implemented on a lightweight platform, and it uses a multi-beam lidar for point cloud registration. In 2020, the same authors updated the LeGO-LOAM framework and came up with LIO-SAM [9], which includes inertial measurement unit (IMU) pre-integration factors and GPS factors. It tightly couples lidar with IMU and realizes trajectory estimation and

map reconstruction with high precision and in real time. In the same year, the algorithm framework LOAM_Livox [10] using solid-state lidar was proposed, and it innovatively uses reflection intensity to remove defects in laser points and improve the robustness of the system. In addition, Google's open-source Cartographer algorithm [11] was a milestone in Lidar SLAM, which originally only supported 2D mapping with a single-line lidar. However, with the new version, 3D laser point cloud reconstruction is also supported. The core idea of the algorithm is to eliminate the accumulated errors in the process of reconstruction using closed-loop detection, and the search method of branch and bound is adopted to greatly increase the speed of the algorithm. However, a multi-beam lidar has a large volume and a high cost. Meanwhile, the point cloud registration algorithm is highly complex and requires a lot of time to reconstruct environmental information. Therefore, 3D reconstruction with a multi-beam lidar cannot be widely promoted in the field of large-scale information acquisition such as in forest areas.

In view of the above problems, single-line lidar has attracted much attention for its light weight and low price, and it has become basic equipment for scene reconstruction. At the same time, two-dimensional environment map reconstruction can be achieved with a single-line lidar, such as GMapping [12] based on the particle filter, Karto SLAM [13] based on pose graph optimization, Hector SLAM [14] based on occupancy grid map matching, and Google Cartographer [11] based on 3D grid map matching. However, single-line lidar mapping can only obtain the 2D information of the plane where the current lidar is located [15], and it cannot obtain spatial 3D information due to its hardware limitations.

With the development of SLAM technology, it is also possible to obtain 3D space point cloud information with a single-line lidar. Hähnel et al. [16] constructed a two-dimensional map with a horizontally placed single-line lidar to recover attitude transformation and a vertically placed single-line lidar to construct point cloud information. In order to collect 3D map information, Zhang et al. [7] installed a single-line lidar on a rotating platform so that the system could have a larger field of view. Zhang et al. [17] used visual odometry to estimate the ego-motion of a single-line lidar, providing a rough estimation of the pose for point cloud registration, and then used the lidar odometry based on scan matching to optimize the pose and register the point cloud. Bosse et al. [18] constructed a Zebedee device, which is composed of a single-line lidar and an inertial measurement unit (IMU). However, the above method relies on a single sensor, which has problems such as a low accuracy, poor robustness, and a complex mechanical structure.

Relying on multi-sensor fusion technology, the above problems can be solved by fusing the laser, vision, IMU, code disc, GPS, and other sensor information [19]. Among them, vision can provide pose estimation in a long-term slow motion state, and IMU can compensate for the positioning deviation during fast motion with its high-frequency measurement advantage. The current fusion of vision and IMU is mainly divided into the following two categories: Kalman filter [20–25] fusion and nonlinear optimization [11,26–29] fusion. Chen et al. [30] used the Kalman filter algorithm to fuse RGB-D images and IMU data to obtain the real-time pose of a mobile robot, and they bounded a single-line lidar to the mobile robot for indoor point cloud reconstruction. However, the Kalman filter can only be optimized based on the data of the previous frame, it cannot obtain the global optimal solution, and the lidar mounted on the wheeled robot platform is not suitable for the scanning of complex terrain in forest areas.

To solve the above problems, in this paper, we propose a real-time reconstruction method of a 3D point cloud based on visual–IMU fusion with a single-line lidar, and we construct a low-cost handheld forest information collection device. The method adopts multi-sensor tightly coupled fusion technology to provide accurate pose estimation for a single-line lidar. Then, we obtain an accurate 3D point cloud map using point cloud projection, densification, and loop closure correction based on pose estimation. Finally, we verify the mapping accuracy and real-time performance through experiments, and the method proposed in this paper can also be seamlessly extended to other devices, such as a multi-beam lidar and a solid-state lidar.

## 2. Real-Time Reconstruction System of Forest Point Cloud

A real-time reconstruction system of a point cloud using a single-line lidar based on visual–IMU fusion in a forest is shown in Figure 1. The whole system consists of a monocular camera, IMU, and a single-line lidar. In order to ensure a uniform density distribution of the 3D point cloud map, the camera is placed horizontally to obtain the front image, and the single-line lidar is placed vertically for longitudinal scanning. When the system moves in the forest area, the monocular camera obtains forest image information and the IMU sensor obtains the angular velocity and acceleration. The global odometry information of the system is obtained by the fusion of the two sensors, and the single-line lidar obtains the laser point cloud plane.

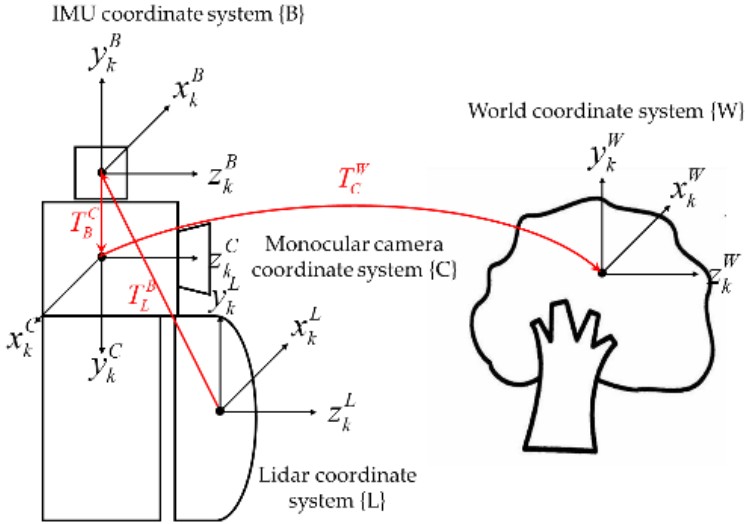

**Figure 1.** Point cloud real-time reconstruction system with a single-line lidar based on visual–IMU fusion.

The overall framework of this paper is shown in Figure 2. First, we use a nonlinear optimization fuse method to fuse the visual reprojection error extracted from the monocular camera images and the IMU pre-integration error from the IMU data, which can project the pose estimation matrix $T_C^W$ from the monocular camera coordinate system (C) to the global world coordinate system (W). Then, by calculating the pose transformation matrix $T_L^W$ of the single-line lidar coordinate system (L) corresponding to (W), the current laser data in the forest can be projected and reconstructed to form a three-dimensional point cloud map. The monocular camera coordinate system (C), the IMU coordinate system (B), and the lidar coordinate system (L) are bound together, and the relative positions of $T_B^C$ and $T_L^B$ between (C), (B), and (L) can be directly obtained through external parameter calibration. Meanwhile, each sensor is time-synchronized.

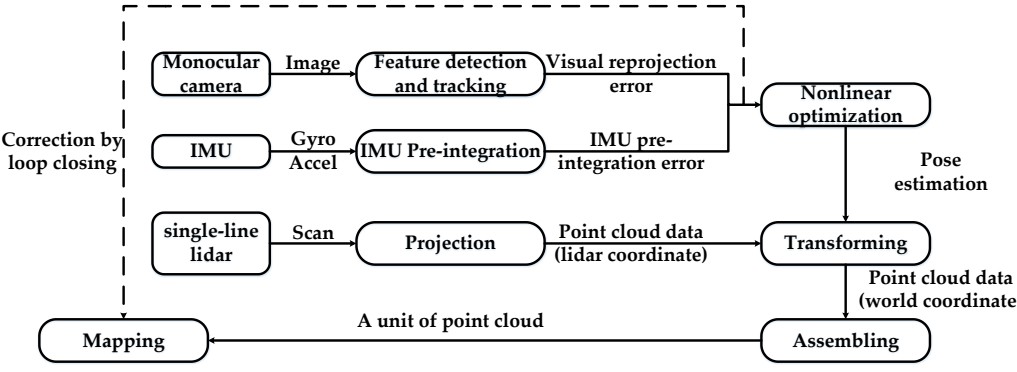

**Figure 2.** The overall framework of forest point cloud real-time reconstruction method with a single-line lidar based on visual–IMU fusion.

### 3. Pose Estimation Based on Visual–IMU Fusion

In order to achieve the pose estimation of the forest point cloud reconstruction system robustly and accurately, we implement a visual–IMU fusion SLAM system based on nonlinear optimization by minimizing the errors from the camera and IMU.

### 3.1. Reprojection Error Based on Vision

The monocular camera collects forest image information and provides the initial image sequence for the forest point cloud reconstruction system. At the same time, in order to avoid the loss of the feature track, this paper establishes the following constraints to define the key frames of the obtained image sequence:

$$\begin{cases} \dfrac{\sum\limits_{k=1} (V_{k+1}-V_k)}{k} > \alpha \\ F_k < \beta \end{cases} \tag{1}$$

where $V_{k+1} - V_k$ is the parallax between the $(k+1)$th frame and the $k$th frame; $F_k$ is the number of feature points of the $k$th frame; and $\alpha$ and $\beta$ are the predefined thresholds. In Formula (1), there are two strategies for key frame selection. The first one is the average parallax of the key frames. If we add one new frame, the average parallax of the tracked features in all key frames exceeds the specified threshold, and we treat the frame as a new key frame. Another strategy is the number of feature points. If the number of tracked feature points is below the specified threshold, the frame is regarded as a new key frame.

This paper uses the KLT sparse optical flow algorithm [31] to extract and track the feature points of image key frames. Therefore, we can form matching feature point pairs between adjacent key frames. At the same time, for each new image, we use the RANSAC algorithm [32] with the basic matrix model for outlier elimination.

In the feature point method, camera motion is often estimated by minimizing the reprojection error based on point pairs from adjacent frames. This paper adopts the difference between the spatial coordinates and the pixel coordinates of the feature points to obtain the visual reprojection error between two adjacent key frames:

$$r_{C_{kp}} = u_{k+1} - \frac{1}{z_k^p} K T_k^w P_k \tag{2}$$

In Equation (2), $P_k = \left( x_k^p, y_k^p, z_k^p \right)^T$ is the 3D spatial coordinate of the $p$th feature point in the $k$th frame; $u_{k+1} = \left( u_{k+1}^p, v_{k+1}^p, 1 \right)^T$ is the pixel coordinate of the $p$th feature point in the $k+1$th frame, expressed in homogeneous coordinates; $T_k^w$ represents the transformation matrix from the $k$th frame to the $k+1$th frame, which is composed of the camera's position and attitude, $p_k^w$ and $q_k^w$, respectively; and $K$ is the camera internal parameter.

The visual reprojection error is obtained from the forest images, and it provides visual constraints for the pose estimation of visual–IMU fusion.

### 3.2. Pre-Integration Error Based on IMU

We collect the angular velocity $\widetilde{\omega}$ and acceleration $\widetilde{a}$ of the IMU sensor on the roll, pitch, and yaw, but they are affected by the gyroscope bias $b_\omega$ and noise $n_\omega$, and the acceleration bias $b_a$ and noise $n_a$. We assume that the derivatives of biases $b_\omega$ and $b_a$ and noises $n_\omega$ and $n_a$ are Gaussian; the real angular velocity $\omega$ and acceleration $a$ where the system is located are given by

$$\omega = \widetilde{\omega} - b_\omega - n_\omega, a = \widetilde{a} - b_a - n_a \tag{3}$$

Several IMU measurements are sampled between the $k$th and $k+1$th frames of the camera, and the position $p_{k+1}$, speed $v_{k+1}$, and attitude $q_{k+1}$ at the $k+1$th frame can be obtained by integrating the IMU measurement values between the two frames. However, the system needs to recalculate the integral between the two frames every time after

the pose of the $k$th frame is optimized, which is time consuming. Therefore, this paper adopts the pre-integration method in VINS-Mono [33] to transform the integration term corresponding to the world coordinate system between the $k$th and $k + 1$th frames into the pre-integration term corresponding to the $k$th frame. At the same time, due to the bias and noise of the IMU pre-integration term, the IMU sensor usually estimates the motion by minimizing the pre-integration error. This paper uses the pre-integration error $r_{I_k}$ of the IMU term. We view the pre-integration component between two frames as the measured value, and all state variables of two frames are subtracted as the observed value:

$$
\begin{bmatrix} r_p \\ r_v \\ r_q \\ r_{ba} \\ r_{b\omega} \end{bmatrix} = \begin{bmatrix} R_w^{b_k}\left(p_{k+1}^w - p_k^w - v_k^w \Delta t + \frac{1}{2}g^w \Delta t^2\right) - \alpha_{k+1}^k \\ R_w^k\left(v_{k+1}^w - v_k^w + g^w \Delta t\right) - \beta_{k+1}^k \\ 2\left[q_k^{k+1} \otimes \left(q_w^k \otimes q_{k+1}^w\right)\right]_{xyz} \\ b_{k+1}^a - b_k^a \\ b_{k+1}^w - b_k^w \end{bmatrix} \tag{4}
$$

where $r$ represents the error of the state quantity; $g$ represents the acceleration of gravity, which is 9.8 m/s$^2$; $\Delta t$ represents the time interval between two adjacent frames $b_k$ and $b_{k+1}$; $\alpha$ and $\beta$ represent the pre-integration term of $p$ and $v$, respectively; and $[\cdot]_{xyz}$ represents the three-dimensional vector consisting of the imaginary part of the quaternion $(x, y, z)$. Therefore, through the above preprocessing, the IMU information is converted into the IMU pre-integration error, which provides IMU constraints for the pose estimation of visual–IMU fusion.

### 3.3. Pose Estimation and Optimization Based on Visual–IMU Fusion

Both the visual odometry and the IMU odometry can obtain the pose transformation matrix. However, the feature point pairs are easily lost when the visual odometry moves rapidly; the IMU data can easily be diverged. Therefore, we adopt a tightly coupled approach to construct a least squares problem of minimizing the visual reprojection error Equation (2) and the IMU pre-integration error Equation (4), and we obtain the pose estimation using nonlinear optimization:

$$
\chi = \underset{\chi}{\operatorname{argmin}} \left\{ \sum_{k \in F} \sum_{p \in P} \left\| r_{C_{kp}} \right\|^2 + \sum_{k \in F} \left\| (r_{I_k}) \right\|^2 \right\} \tag{5}
$$

$$
\chi = \{p_k^w, v_k^w, q_k^w, b_k^w, b_k^a\} \tag{6}
$$

where $F$ represents all image frames; $P$ represents the feature points extracted from all image frames; $p_k^w$, $v_k^w$, and $q_k^w$ represent the position, velocity, and attitude, respectively, of the $k$th key frame under the world coordinate system (W); and $b_k^w$ and $b_k^a$ represent the bias of the gyroscope and the acceleration at the $k$th key frame, respectively. Finally, we adopt the Gauss–Newton method to solve Equation (5) in order to obtain poses $p_k^w$ and $q_k^w$ at the $k$th key frame. According to the pose information of the previous frames, the global pose transformation matrix $T_C^W$ of the current time can be obtained.

Therefore, this paper obtains the global real-time positioning information of the forest point cloud reconstruction system at the current time.

## 4. The 3D Point Cloud Reconstruction

We can obtain global real-time pose estimation through visual–IMU fusion. In order to form a 3D point cloud map, we adopt projection, densification, and loop closure correction to combine laser data with pose information.

*4.1. The 3D Point Cloud Space Projection*

The single-line lidar can simultaneously obtain the angle and distance $(\theta, d)$ of each laser point data in the lidar polar coordinate. By taking into account the synchronization between the lidar data and the odometry information, this paper only receives the lidar data at the time of the $k$th visual key frame of the camera and maps them to the world coordinates according to the real-time positioning information.

Firstly, the angle and distance $(\theta_k^i, d_k^i)$ measured in the lidar polar coordinate is directly projected to the lidar coordinate ($L$):

$$\begin{pmatrix} X_k^{i(L)} \\ Y_k^{i(L)} \\ Z_k^{i(L)} \end{pmatrix} = d_k^i \begin{pmatrix} \cos \theta_k^i \\ \sin \theta_k^i \\ 0 \end{pmatrix} \tag{7}$$

where $d_k^i$ indicates the distance of the $i$th laser point during the $k$th scan; $\theta_k^i$ is the corresponding angle; and the superscript ($L$) indicates the lidar coordinate.

As the single-line lidar is placed vertically, the laser point data produce distortion errors along with the system movement, resulting in a spiral deflection. In this paper, IMU information is used to eliminate spiral deflection. During the laser point scanning at the $k$th frame, we obtain poses $p_s$ and $p_e$ though IMU integration at the beginning and the end of the lidar rotation. This paper adopts linear interpolation to eliminate spiral deflection, and the pose $p_i$ of each intermediate point can be approximated as

$$p_i = \frac{(e - i)p_s + (i - s)p_e}{e - s} \tag{8}$$

where $s$ and $e$ represent the time stamps at the beginning time and the end time, respectively, and $i$ represents the time stamp of the middle laser point of the single circle scanning. $p_s$, $p_e$, and $p_i$ represent the system poses at the time stamps of $s$, $e$, and $i$, respectively.

Then, each laser point is adjusted to form the correct coordinates of the laser point:

$$\begin{pmatrix} X_k^{i'(L)} \\ Y_k^{i'(L)} \\ Z_k^{i'(L)} \end{pmatrix} = p_i^T \begin{pmatrix} X_k^{i(L)} \\ Y_k^{i(L)} \\ Z_k^{i(L)} \end{pmatrix} \tag{9}$$

Then, the pose estimation $T_C^W$ of the key frame is obtained using tightly coupled multi-sensor fusion as mentioned earlier. According to the relative pose transformation between multiple sensors, $T_B^C$ and $T_L^B$, we obtain the global pose transformation matrix $T_L^W$ of the forest point cloud reconstruction system, which transforms laser points in the lidar coordinate $(X_k^{i'(L)}, Y_k^{i'(L)}, Z_k^{i'(L)})$ to the world coordinate $(X_k^{i(W)}, Y_k^{i(W)}, Z_k^{i(W)})$. Superscript ($W$) indicates that the point is in the world coordinate system:

$$T_L^W = T_C^W T_B^C T_L^B \tag{10}$$

$$\begin{pmatrix} X_k^{i(W)} \\ Y_k^{i(W)} \\ Z_k^{i(W)} \end{pmatrix} = T_L^W \begin{pmatrix} X_k^{i'(L)} \\ Y_k^{i'(L)} \\ Z_k^{i'(L)} \end{pmatrix} \tag{11}$$

According to (11), the single-line lidar data can be projected in real time and converted to the world coordinate system to form a 3D point cloud at the key frame.

### 4.2. The 3D Space Point Cloud Densification

In order to align the data of the visual frame and the lidar frame, we only collect point cloud data at the visual key frame, which causes data loss in the adjacent two frames $k$ and $k + 1$.

In order to obtain a dense point cloud map, this paper utilizes the local features of point clouds at two adjacent frames to fill the point cloud information between two frames. As shown in Figure 3, we obtain the pose matrixes $T_k$ and $T_{k+1}$ at the $k$th and $k + 1$th frames, respectively. The pose matrix includes a rotation matrix $R$ and a translation vector $t$. In order to provide the pose for each moment between two frames, we adopt different interpolation strategies to interpolate $R$ and $t$. For the rotation matrix $R$, the pose is often expressed by the quaternion, so we use nonlinear spherical interpolation (Equation (13)), while for the translation vector $t$, we use linear interpolation (Equation (14)). By combining the pose of each moment with the lidar data of the world coordinate system $(X_k^{i(W)}, Y_k^{i(W)}, Z_k^{i(W)})$, the loss of 3D point cloud data can be filled:

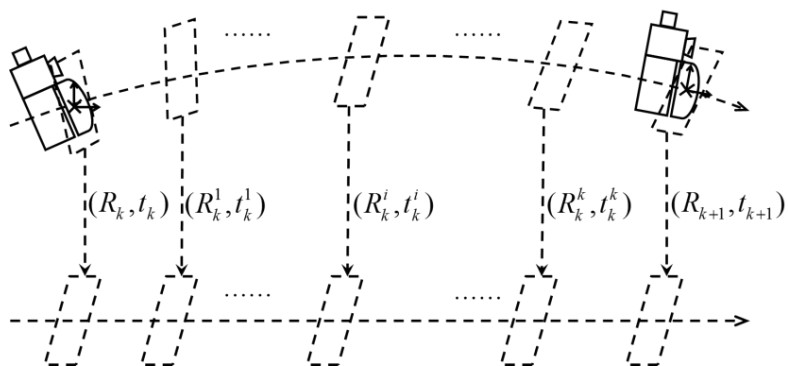

**Figure 3.** The 3D point cloud densification model.

$$\begin{pmatrix} X_l^{i(W)} \\ Y_l^{i(W)} \\ Z_l^{i(W)} \end{pmatrix} = R_k^l \begin{pmatrix} X_k^{i(W)} \\ Y_k^{i(W)} \\ Z_k^{i(W)} \end{pmatrix} + t_k^l \tag{12}$$

$$R_k^l = \frac{\sin\left(\theta_k - \theta_k^l\right) R_k + \sin(\theta_k^l) R_{k+1}}{\sin \theta_k} \tag{13}$$

$$t_k^l = \frac{(n - l) t_k + l t_{k+1}}{n} \tag{14}$$

where $R_k$ represents the rotation matrix of the system at the initial moment of the $k$th key frame; $t_k$ is the translation vector of the system at the initial moment of the $k$th key frame; $n$ is the total number of point cloud rings to be filled between two key frames; and $l$ indicates the sequence of point cloud rings between two key frames. $R_k^l$ is obtained by using nonlinear spherical interpolation, and $t_k^l$ is obtained by using linear quadratic interpolation.

The sparse point cloud at the key frame is densified between frames to form the initial point cloud map.

### 4.3. The 3D Space Point Cloud Loop Closure Correction

Since the pose estimation of multi-sensor fusion is a recursive process, the accumulated error will cause the overall deviation of the 3D reconstruction. This section adopts loop closure detection and optimization based on the visual key frame to improve the accuracy of the mapping.

In the process of 3D reconstruction, it is necessary to continuously detect whether the current frame is similar to a historical frame in order to incorporate a loop. Due to the huge number of key frames in large-scale forest scenes, it is impossible to compare the current

frame with all previous key frames one by one. Therefore, this paper adopts the DBoW2 model with bags of binary words [34] to form a KD-Tree of feature points. If the current frame $K_{cur}$ meets the consistency detection with a past frame, the past frame is regarded as a closed-loop frame $K_{loop}$.

After the closed-loop frame is detected, the accurately matched feature point pair is established between the current frame $K_{cur}$ and the closed-loop frame $K_{loop}$. Meanwhile, we obtain the poses of $K_{cur}$ and $K_{loop}$. According to the pose optimization proposed in this paper in Section 3.3, we use the reprojection error of feature point pairs and the IMU pre-integration error between $K_{cur}$ and $K_{loop}$ as constraints in Equation (5). The correction matrix $T_{cur}^{loop}$ can be obtained by tightly coupled optimization. Therefore, the 3D point cloud data at the current frame $K_{cur}$ are corrected to

$$
\begin{pmatrix}
X_{cur(cor)}^{i(W)} \\
Y_{cur(cor)}^{i(W)} \\
Z_{cur(cor)}^{i(W)}
\end{pmatrix}
= T_{cur}^{loop}
\begin{pmatrix}
X_{cur}^{i(W)} \\
Y_{cur}^{i(W)} \\
Z_{cur}^{i(W)}
\end{pmatrix}
\tag{15}
$$

The above process only corrects the point cloud information of the current frame $K_{cur}$, but the point cloud information between $K_{loop}$ and $K_{cur}$ still has a cumulative error.

We define one frame $K_j$ between $K_{loop}$ and $K_{cur}$, assuming that we know the pose of $K_j$ through the pose estimation of visual–IMU fusion. This paper establishes the residual function of the pose error. It includes two parts. The first one is caused by the translation and yaw angle between the $K_j$th key frame and the adjacent $K_{j+1}$th key frame; the other one is caused by $T_{cur}^{loop}$:

$$
r_j^{j+1} =
\begin{bmatrix}
R(\hat{\phi}_j, \hat{\theta}_j, \psi_j)^{-1}(t_{j+1} - t_j) - \hat{t}_j^{j+1} \\
\psi_{j+1} - \psi_j - \hat{\psi}_j^{j+1}
\end{bmatrix}
\tag{16}
$$

$$
r_{cur}^{loop} =
\begin{bmatrix}
t_{cur} - t_{loop} - \hat{t}_{cur}^{loop} \\
\psi_{cur} - \psi_{loop} - \hat{\psi}_{cur}^{loop}
\end{bmatrix}
\tag{17}
$$

In Equations (16) and (17), $R(\hat{\phi}_j, \hat{\theta}_j, \psi_j)$ represents the rotation matrix that only optimizes the yaw angle, and $t$ represents the translation matrix; $\phi$, $\theta$, and $\psi$ represent the roll angle, pitch angle, and yaw angle, respectively. Here, the superscript $\wedge$ represents the accurate value, which does not participate in the optimization process; $\hat{t}_j^{j+1}$ represents the translation vector obtained by the pose estimation of the multi-sensor fusion between the $j$th and $j + 1$th frames, and $\hat{\psi}_j^{j+1}$ is the yaw angle obtained from the rotation matrix in the pose estimation; and $\hat{t}_{cur}^{loop}$ and $\hat{\psi}_{cur}^{loop}$ are obtained by the correction matrix $T_{cur}^{loop}$. This paper constructs the loss function of the loop closure optimization by using the translation matrix $t$ and the yaw angle $\psi$ as the optimization variables:

$$
\min_{t, \psi} \left\{ \sum_{j, j+1 \in S} \left\| r_j^{j+1} \right\|^2 + \sum_{cur, loop \in L} \left\| r_{cur}^{loop} \right\|^2 \right\}
\tag{18}
$$

where $S$ is the set of adjacent frames, which are taken as constraints, and $L$ is the set of closed-loop frames, which are taken as constraints. The correction matrix $T_{j(cor)}$ of each frame can be obtained by minimizing the loss function using the Gauss–Newton method. Moreover, the 3D point cloud, which has a cumulative error, can be adjusted to

$$
\begin{pmatrix} X^{i(W)}_{j(cor)} \\ Y^{i(W)}_{j(cor)} \\ Z^{i(W)}_{j(cor)} \end{pmatrix} = T_{j(cor)} \begin{pmatrix} X^{i(W)}_{j} \\ Y^{i(W)}_{j} \\ Z^{i(W)}_{j} \end{pmatrix}
\tag{19}
$$

Finally, this paper generates globally consistent 3D point cloud maps.

## 5. Experiments and Discussion

### 5.1. Experimental System Design

This paper constructs a forest point cloud real-time reconstruction system, as shown in Figure 4. The system includes a monocular camera, IMU, a hand-held bracket, a single-line lidar, and a computing platform. The monocular camera and the IMU use a realsense D435i device. The single-line lidar uses Rplidar A2, which has a 360° field of view and a 0.9° angular resolution. Each sensor is time-synchronized in advance, and the transformations among sensors are regarded as constant values, which are calibrated offline. The computing device adopts a PC with a quad-core 2.5 GHz frequency and an i7 processor, which uses the Ubuntu18.04 operating system and the ROS robot operating system.

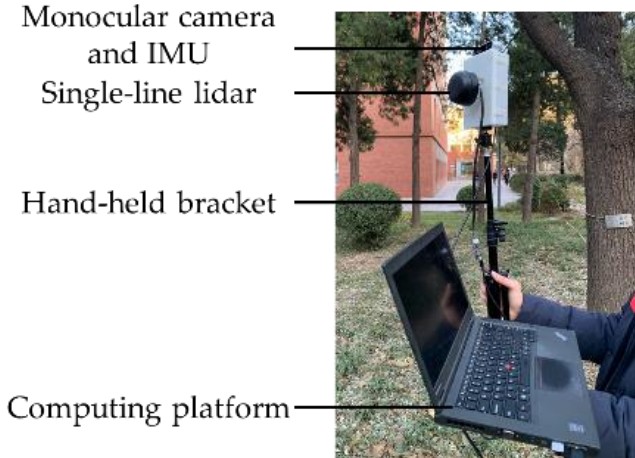

**Figure 4.** The 3D point cloud reconstruction system.

We collect data from the forest of Dongsheng Bajia Country Park in Haidian District of Beijing with the designed real-time reconstruction system of the forest point cloud. The experiments regarding point cloud densification and loopback correction are carried out separately. In addition, we conduct studies at different scenes, such as a single tree, a row of trees, and some forest areas. In the experiment, the monocular camera outputs forest image information at a frequency of 20 Hz, the IMU outputs acceleration and angular velocity information at a frequency of 200 Hz, and the single-line lidar outputs forest laser point data at a frequency of 12 Hz.

### 5.2. The 3D Point Cloud Reconstruction in Forest Area

During the 3D point cloud reconstruction process, the lidar data are first projected to generate a sparse point cloud according to Figure 5a, and then the sparse point cloud is densified into a 3D point cloud map, as shown in Figure 5b. When the system returns to the starting point, a loop is formed based on key frame detection. As shown in Figure 5c, the 3D point cloud of the forest area realizes loop correction and reduces the cumulative error. We generate a globally consistent 3D point cloud map, where features such as ground, fences, and trees can be clearly seen.

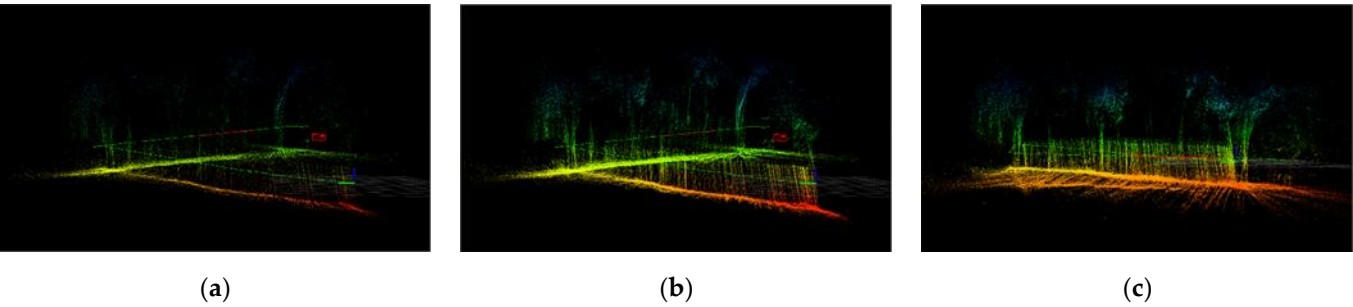

| (**a**) | (**b**) | (**c**) |

**Figure 5.** The 3D point cloud densification and loop closure correction: (**a**) sparse point cloud; (**b**) dense point cloud; (**c**) dense point cloud loop closure correction.

In order to analyze the robustness and availability of the reconstruction method proposed in this paper with a single-line lidar, this paper collects data from forest areas with undulating terrain and a messy arrangement of trees, and a 3D point cloud map is obtained. The reconstruction map is shown in Figure 6. It can be seen that, in the area with rich environmental features, the 3D point cloud is well preserved and tree reconstruction is more intensive. In the area with sparse environmental features, localization also has good convergence, and the 3D point cloud can still accurately reflect environmental information. Bushes, trees, branches, leaves, and paths all achieve a good reconstruction effect.

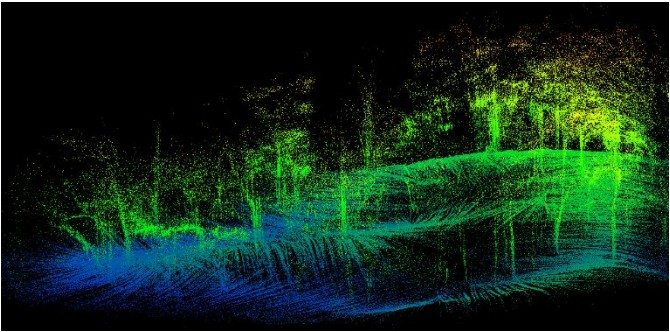

**Figure 6.** The 3D forest point cloud map of a complex scene.

*5.3. Experiment Comparison of Different Devices*

In order to evaluate the local accuracy of the 3D point cloud map, this paper compares it with manual meter scale measurements and a RoboSense RS-Ruby multi-beam lidar (80-line lidar) based on the LIO-SAM framework in three different scenes. LIO-SAM, the mainstream of current reconstruction methods, is obviously superior to other methods in the field of outdoor reconstruction [9,35,36]. Figure 7 shows several maps of a single tree, a row of trees, and a part of the forest area using different devices. This paper takes the measurement values from the manual meter scale as the real values $D$ and the measurement values from the two other devices as the measurement results $D_i$. The absolute errors caused by the difference between $D$ and $D_i$, compared with the real values, are taken as the relative errors of the construction of mapping $\delta$:

$$\delta = \frac{|D_i - D|}{D} \times 100\% \tag{20}$$

In the measurement of a single tree, this paper collects the height and the diameter of the tree at 1.25 m, 1.58 m, and 1.86 m from the root of the ground using the different devices. The measurement data are shown in Table 1. Analyzing the data shows that the relative error of the method proposed in this paper is 3.66%, and the relative accuracy can reach 96.34%.

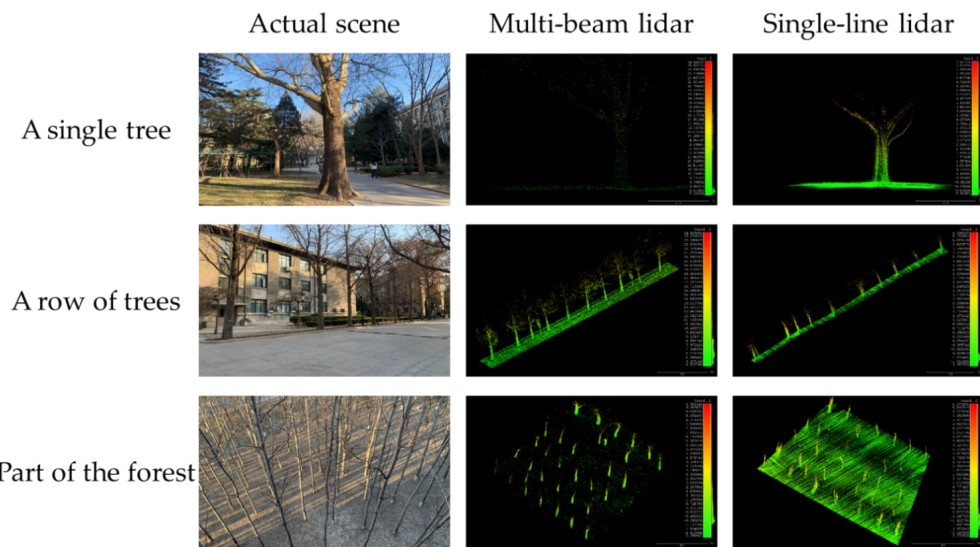

**Figure 7.** Accuracy comparison of 3D point cloud reconstruction.

**Table 1.** Measurement data of a single tree.

| A Single Tree | The Diameter at 1.25 m | | The Diameter at 1.58 m | | The Diameter at 1.86 m | | The Height | | Average Error |
|---|---|---|---|---|---|---|---|---|---|
| | $D_i$ (m) | $\delta$ (%) | $D_i$ (m) | $\delta$ (%) | $D_i$ (m) | $\delta$ (%) | $D_i$ (m) | $\delta$ (%) | $\overline{\delta}$ (%) |
| Real value | 1.16 | 0 | 1.13 | 0 | 1.06 | 0 | 4.38 | 0 | 0 |
| Multi-beam lidar | 1.18 | 1.72 | 1.11 | 1.77 | 1.04 | 1.89 | 4.45 | 1.60 | 1.75 |
| Single-line lidar | 1.20 | 3.45 | 1.17 | 3.54 | 1.02 | 3.77 | 4.21 | 3.88 | 3.66 |

Diameter at breast height (DBH) is the most direct factor reflecting tree growth. In order to further evaluate the measurement accuracy of DBH, this paper adopts circle fitting [37] to extract the tree diameter, as shown in Figure 8a. For better DBH estimates, 10 cm-thick sections are selected at heights of 1.25 m, 1.58 m, and 1.86 m, as shown in Figure 8b. We measure different DBH values of 20 trees in the forest area and preprocess the data to eliminate outliers. We use the RMSE, bias, and Pearson's *r* parameters for accuracy assessment. The results are shown in Table 2. The average RMSE of DBH is 4.35 cm in the reconstruction with the single-line lidar compared with the real values, which is 2 cm different from that of the multi-beam lidar. The bias values for the DBH estimations are also presented in Table 2. They are 0.476 cm and −0.273 cm for the multi-beam lidar and single-line lidar, respectively. The average values close to zero show that the DBH estimations are almost unbiased. Pearson's *r* represents the correlation between the estimated and the true values of DBH. It is obvious that they all have a high correlation.

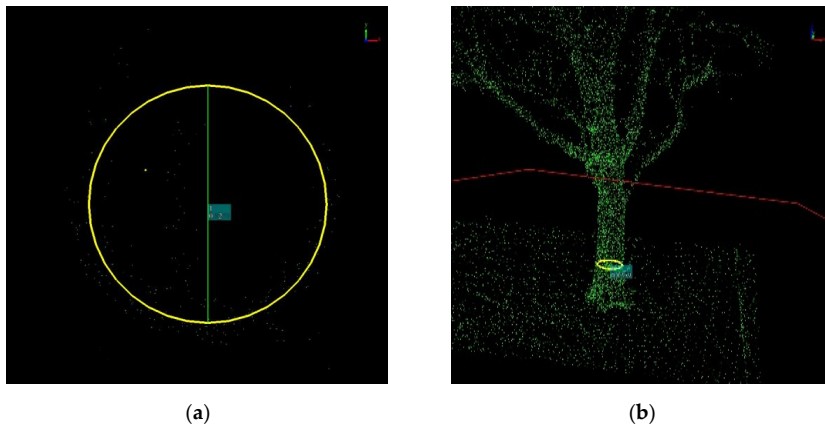

| (a) | (b) |

**Figure 8.** Cross-sections of point cloud maps: (**a**) circle fitting; (**b**) 10 cm-thick sections.

**Table 2.** Accuracy assessment for DBH parameter.

| Height (m) | Different Devices | | | | | |
|---|---|---|---|---|---|---|
| | Multi-Beam Lidar | | | Single-Line Lidar | | |
| | RMSE (cm) | Bias (cm) | Pearson's *r* | RMSE (cm) | Bias (cm) | Pearson's *r* |
| 1.25 | 2.92 | 0.439 | 0.987 | 4.47 | −0.865 | 0.947 |
| 1.58 | 2.47 | 0.620 | 0.987 | 4.08 | 0.013 | 0.945 |
| 1.86 | 1.69 | 0.370 | 0.984 | 4.49 | 0.016 | 0.938 |
| Average | 2.36 | 0.476 | 0.986 | 4.35 | −0.273 | 0.944 |

For tree spacing in a large-scale forest area, this paper selects a row of 10 trees with different spacings to measure the distance between adjacent trees. The results are shown in Table 3, in which it can be seen that the method proposed in this paper constructs a 3D point cloud of a row of trees; the relative error of the tree spacing is 4.65%, and the relative accuracy can reach 95.35%.

**Table 3.** Measurement data of a row of trees.

| Tree Spacing | 1–2 | 2–3 | 3–4 | 4–5 | 5–6 | 6–7 | 7–8 | 8–9 | 9–10 | Average Error |
|---|---|---|---|---|---|---|---|---|---|---|
| | $D_i$ (m) | $D_i$ (m) | $D_i$ (m) | $D_i$ (m) | $D_i$ (m) | $D_i$ (m) | $D_i$ (m) | $D_i$ (m) | $D_i$ (m) | $\bar{\delta}$ (%) |
| Real value | 6.30 | 3.27 | 3.75 | 4.80 | 4.80 | 4.25 | 3.80 | 4.36 | 5.13 | 0 |
| Multi-beam lidar | 6.50 | 3.33 | 3.85 | 5.10 | 4.80 | 4.27 | 3.90 | 4.32 | 5.10 | 2.06 |
| Single-line lidar | 6.30 | 3.01 | 3.55 | 5.22 | 4.66 | 4.20 | 4.02 | 4.65 | 5.30 | 4.65 |

At the same time, this paper also selects a trapezoidal forest area for study and measures its short base, long base, altitude, lateral side, and the number of trees. The coverage area is approximated by the trapezoidal area. By analyzing the data in Table 4, it can be seen that the relative error of the method proposed in this paper is 1.92% in part of the forest, and the relative accuracy can reach 98.08%.

**Table 4.** Measurement data of some forest areas.

| Part of the Forest | Short Base | | Long Base | | Altitude | | Lateral Side | | Average Error | Coverage Area | Number |
|---|---|---|---|---|---|---|---|---|---|---|---|
| | $D_i$ (m) | $\delta$ (%) | $D_i$ (m) | $\delta$ (%) | $D_i$ (m) | $\delta$ (%) | $D_i$ (m) | $\delta$ (%) | $\bar{\delta}$ (%) | (m$^2$) | |
| Real value | 25.3 | 0 | 35.8 | 0 | 24.5 | 0 | 26.6 | 0 | 0 | 748.47 | 32 |
| Multi-beam lidar | 24.9 | 1.58 | 35.2 | 1.68 | 24.9 | 1.63 | 27.1 | 1.88 | 1.69 | 748.25 | 32 |
| Single-line lidar | 24.8 | 1.98 | 35.3 | 1.40 | 25.0 | 2.04 | 27.2 | 2.26 | 1.92 | 751.25 | 32 |

As shown in Figure 7, the 3D point cloud maps constructed using different sensors are compared with the actual scene, and they can completely reflect the abundant environmental information in the forest areas. Furthermore, it can be seen from the data in the above four tables that the algorithm proposed in this paper has good measurement results for tree height, tree diameter at breast height, and tree spacing and distribution in forest areas. Moreover, the measured average relative error of the algorithm in this paper is less than 5% compared with real environmental values, while the measured average relative error of the multi-beam lidar is about 2%. The accuracy of the maps created using the two methods is very similar, but the cost of the proposed algorithm is less than one-tenth of that of the multi-beam lidar. In addition, the point cloud registration algorithm based on the LIO-SAM of the multi-beam lidar has high complexity. It takes more time compared with real-time reconstruction based on the visual–IMU fusion of a single-line lidar proposed in this paper. Considering that forest development pays more attention to the distribution of trees, while the reconstruction of branches and leaves are not key factors, the algorithm proposed in this paper achieves better real-time performance and lower price at the cost of losing parts of branches and leaves in the point cloud data.

### 5.4. Experimental Comparison of Different Methods

In the experimental comparison of the different methods with the same device, this paper uses the Kalman filter algorithm [19] and the nonlinear optimization algorithm proposed in this paper to collect point cloud data from the same forest area, and the reconstruction results are shown in Figure 9.

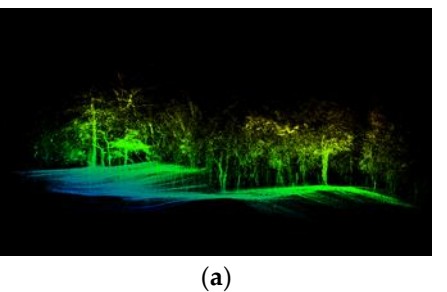
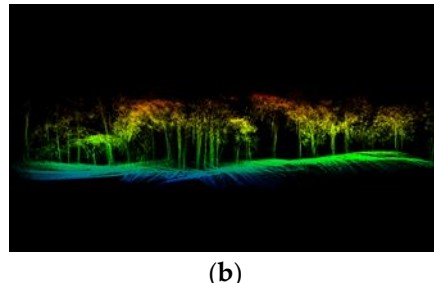

| (**a**) | (**b**) |

**Figure 9.** Comparison of reconstruction results of different methods: (**a**) 3D reconstruction based on Kalman filter; (**b**) 3D reconstruction based on the method proposed in this paper.

At the initial stage of the experiment, both methods construct the 3D scene map of the forest area well, as shown in the right part of Figure 9a,b. With an increase in the amount of collected data, a scene shift occurs with the Kalman filter method. However, with the method proposed in this paper, we can still accurately collect the terrain data, as shown in the left part of Figure 9a,b. In this paper, the root mean square error is used as the accuracy evaluation standard between the point cloud maps. The experiments show that the proposed method reduces the root mean square error by 4.65% compared with the method based on the Kalman filter.

In this paper, the time cost of each part in the point cloud processing is calculated as the evaluation index of the 3D reconstruction system, as shown in Table 5. The Kalman filter algorithm in the original literature iterates with the pose covariance matrix, and its calculation amount increases by the order of magnitude. Moreover, it is only optimized based on the pose of the last frame, lacking loopback and unable to eliminate cumulative error. Compared with the 3D reconstruction based on the Kalman filter, the method proposed in this paper has strong robustness and high precision, and its time cost meets the real-time requirements of a 3D point cloud reconstruction system.

**Table 5.** Average time cost.

| Methods | Time Cost (μs) | | |
|---|---|---|---|
| | Transform to the World Frame | Fill Point Cloud | Loop Elimination Error |
| Kalman filter | 147 | 951 | / |
| Nonlinear optimization | 158 | 571 | 174 |

### 6. Conclusions

In this paper, we propose a forest point cloud real-time reconstruction method with a single-line lidar based on visual–IMU fusion, and we construct a low-cost point cloud acquisition device consisting of a monocular camera, IMU, and a single-line lidar. We adopt a nonlinear optimization method to obtain the pose estimation and then transform laser data using projection, densification, and loop closure correction to form a 3D point cloud map. Meanwhile, we create 3D point cloud maps of a single tree, a row of trees, and part of a forest region. The experimental results show that the average relative error of the proposed method is 3.41% compared with real values, and the root mean square error is reduced by 4.65% compared with the Kalman filter fusion algorithm. The average time cost is 903μs and reaches the real-time requirement. The method in this paper is optimized based on all historical data, which makes it easier to achieve the optimum.

**Author Contributions:** Methodology, C.Y.; software, C.Y. and K.L.; validation, C.Y. and K.L.; investigation, C.Y.; data curation, C.Y.; writing—original draft preparation, C.Y.; writing—review and editing, J.Z. and C.H.; visualization, C.Y. and K.L.; project administration, J.Z.; funding acquisition, J.Z. and C.H. All authors have read and agreed to the published version of the manuscript.

**Funding:** This study was financially supported by the National Natural Science Foundation of China (Grant No. 61703047).

**Institutional Review Board Statement:** Not applicable.

**Informed Consent Statement:** Not applicable.

**Data Availability Statement:** Data available on request due to restrictions e.g., privacy or ethical. The data presented in this study are available on request from the corresponding author. The data are not publicly available due to [forest resource protection].

**Conflicts of Interest:** The authors declare no conflict of interest.

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
