# Peer review of "A Forest Point Cloud Real-Time Reconstruction Method with Single-Line Lidar Based on Visual–IMU Fusion"

_applsci, doi:10.3390/app12094442_

Round 1

Reviewer 1 Report

The authors presented a reconstruction method using Lidar. However, I think they should consider the following suggestion to improve the overall state of the manuscript:

  1. They only discussed the usage of a single lidar application without discussing other available methods for reconstruction.
  2. Perhaps other available methods for reconstruction can be the comparison results against the proposed method.
  3. The formatting of the manuscript should be addressed. Some of the sentences are single spacing and some of them are double spacing.
  4. The experiments conducted are quite brief where only three scenes are considered. Perhaps more comprehensive experiments should be considered by the authors.
  5. Some statistical study should be done to see the significance of the results acquired.

Reviewer 2 Report

This is a review report of the manuscript entitled "A Forest Point Cloud Real-time Reconstruction Method with Single-line Lidar Based on Visual-imu Fusion, " submitted to the Applied Sciences journal.

The authors have selected an interesting topic, and it has a good match for the journal scope. However, I request to see some improvements in the manuscript.

  • P3L116: what is LK? Or do authors mean KLT? According to the reference paper [28], the paper proposed Kanade, Lucas and Tomasi have proposed a feature selection and robust tracking algorithm (KLT).
  • In equation 8, the definitions of the used symbols should be added to the text.
  • P6L209: different interpolation strategies, which interpolation method was used. Please mention it.
  • However, until Figure 6, the theoretical part is very well described in the experimental section. There are new experiments that are required. Figure 6, the single tree is almost double size compared with multibeam lidar. In this case, it brings complexity. I should recommend it here; authors must prove it. When table 1 is inspected, authors write only 4 cm! Difference.
  • What is the measurement technique of diameter (DBH) detection in point cloud?
  • Please add cross sections for single tree data, indicated in Table 1 values. Please add two or more-point cloud data in the same point cloud data and take a slice above ground level (1.25m) to understand the deviations.
  • I also suggest circle fitting for tree diameter extraction for precise extraction of geometry. (10.5552/crojfe.2021.1096, this paper may help about it)

Overall, I suggest major revision. After the revision, the correction of amendments will determine to make my final decision.

Round 2

Reviewer 2 Report

The authors have made important improvements. In the current form, it can be acceptable.